# Evaluation of the Impacts of Abandoned Mining Areas: A Case Study with Benthic Macroinvertebrate Assemblages

**DOI:** 10.3390/ijerph182111132

**Published:** 2021-10-22

**Authors:** Mi-Jung Bae, Jeong-Ki Hong, Eui-Jin Kim

**Affiliations:** Freshwater Biodiversity Research Bureau, Nakdonggang National Institute of Biological Resources, Sangju 37242, Korea; tomasx@nnibr.re.kr (J.-K.H.); ejkim@nnibr.re.kr (E.-J.K.)

**Keywords:** abandoned mining area, self-organizing map, network analysis, indicator species analysis

## Abstract

Mining activities are among the most long-lasting anthropogenic pressures on streams and rivers. Therefore, detecting different benthic macroinvertebrate assemblages in the areas recovered from mining activities is essential to establish conservation and management plans for improving the freshwater biodiversity in streams located near mining areas. We compared the stability of benthic macroinvertebrate communities between streams affected by mining activities (Hwangjicheon: NHJ and Cheolamcheon: NCA) and the least disturbed stream (Songjeonricheon: NSJ) using network analysis, self-organizing map, and indicator species analysis. Species richness was lowest at sites where stream sediments were reddened or whitened due to mining impacts in NHJ and NCA. Among functional feeding groups, the ratio of scrapers was lower (i.e., NHJ) or not observed (i.e., NCA) in the affected sites by mining. The networks (species interactions) were less connected in NHJ and NCA than in NSJ, indicating that community stability decreased in the area affected by mining activity. We identified five groups based on the similarity of benthic macroinvertebrate communities according to the gradients of mining impacts using a self-organizing map. the samples from the reference stream (clusters 1 and 5), sites located near the mining water inflow area (cluster 4), sites where stream sediments acid-sulfated (cluster 2), and sites that had recovered from mining impacts (cluster 3). Among the 40 taxa selected as indicators defined from the five clusters in self-organizing map, only few (*Physa acuta*, *Tipula* KUa, and *Nemoura* KUb) indicator species were selected in each cluster representing the mining-impacted sites. Our results highlighted that the benthic macroinvertebrate community complexity was lower in streams affected by mining activity. Furthermore, the range of disturbed areas in the streams, where conservation and management plans should be prioritized, can be quantified by examining alterations in the benthic macroinvertebrate community.

## 1. Introduction

Freshwater ecosystems are most vulnerable to anthropogenic disturbances worldwide [1,2]. Hence, understanding the patterns and drivers of biodiversity loss is essential for foreseeing the changes of freshwater ecosystems to environmental alteration and establishing conservation strategies [3]. Among the existing anthropogenic pressures (e.g., industrialization, urbanization, and land use changes), mining activities have the most long-lasting effects on streams and rivers [4]. In streams and rivers near mining areas, physical and chemical factors are vulnerable to changes, such as organic matter breakdown [5], conductivity increase [6], or sediment contamination [7,8], erosion or deposition [9]. In particular, drainage water from abandoned mining areas (or active and historic mining operations) is the main input source of heavy metal pollution in adjacent streams [10,11]. Heavy metal concentration in water often exceeds the permissible limits recommended for drinking or agricultural use, and such toxic contaminants are destined to severely damage important resources in freshwater after exposure [12]. Despite the permissible limit for water quality criteria, accumulated heavy metal in streambeds due to past mining activities cannot be removed for several decades [13]. Thus, freshwater ecosystems in streams and rivers near mines or abandoned mining areas are disturbed or even destroyed, resulting in the general modification of the composition and functional attributes of aquatic organisms as well as reduced species richness [14].

Benthic macroinvertebrates play an important role in aquatic food webs [7,15]. Because of their high sensitivity to various contaminants, benthic macroinvertebrates have been generally applied to assess the ecological impacts from various anthropogenic disturbances. Benthic macroinvertebrates are closely related to streambed substrate, which provides a place to rest, shelter, and feed [16] and are the essential material exchangers crossing the sediment-water interface [7,16,17]. Therefore, benthic macroinvertebrates are generally known to be the sensitive organisms to the effects of mining activities [12]; identifying and characterizing alterations of benthic macroinvertebrate communities in streams affected by mining activities is useful for detecting the impacts of abandoned mining areas on aquatic ecosystems.

In general, anthropogenic pressures, such as mining impacts, can cause abrupt changes in the composition of functional feeding groups (FFGs) as well as the structure of the benthic macroinvertebrate community [18,19]. For example, the abundance and diversity of species generally decrease in streams located near abandoned mining areas (e.g., [20,21,22,23,24,25,26]). High levels of cadmium and zinc concentration had a negative effect especially on the number of Plecoptera and Trichoptera [27,28]. In particular, the number of sensitive taxa (e.g., stoneflies, mayflies, caddisflies, etc.) sharply decrease, whereas the tolerant taxa (e.g., chironomids, beetles, etc.) become dominant [12,29,30]. Additionally, interactions between FFGs also vary as the available habitat for stream fauna is reduced, the quality of food resources deteriorates, and metal precipitates cause chemical weathering of the streambed [12,31,32]. Consequently, the ratio of scrapers in FFGs, including Psephenidae and Heptageniidae, which usually attach their bodies and scrape algae in boulders and cobbles [33], can rapidly decrease in the affected ecosystem.

To date, when evaluating the changes in stream environments near abandoned mining areas, only the water quality variables or the coarse level of benthic macroinvertebrate data (i.e., family level) have been used (but also see [34,35]), and descriptive or multivariate analyses have been applied to interpret the results. Network analysis [36], which has now been increasingly used to understand the interactions and stability of complex ecological communities, can provide undetectable insights compared with traditional analytical methods that analyze the species separately [37,38]. A self-organizing map [39], which is an unsupervised neural network, is a powerful tool to interpret the relation between ecological community and environment based on visualizing, grouping, and predicting the complex ecological data. The application of self-organizing map has been recently increased especially in the interpretation of the ecological data (e.g., [3,38,40,41]). However, there have been no studies that applied both techniques simultaneously to evaluate the mining impacts on the benthic macroinvertebrate community.

This study aimed to investigate the changes in benthic macroinvertebrate communities in streams near mining areas using modelling approaches (network analysis and self-organizing map). First, we compared the stability of benthic macroinvertebrate communities between two streams within an area affected by mining activities and a reference stream in the least-disturbed area. We applied network analysis to compare the community complexity (the number of interactions among species) among three streams (reference stream and two streams near the mining area). Second, the range of disturbed areas in the streams was quantified by examining alterations in the benthic macroinvertebrate community. We used the self-organizing map, which can classify clusters based on the similarity of community composition to evaluate if there were significant differences of the communities between streams and detect the sites impacted or recovered from mining areas.

## 2. Materials and Methods

### 2.1. Study Area

To compare the impacts of abandoned mining areas on benthic macroinvertebrate communities, we monitored the streams near abandoned mining areas (Hwangjicheon: NHJ and Cheolamcheon: NCA) and in a less disturbed area (Songjeonricheon: NSJ). The research area (Taebaek, Korea) presented Monsoon-influenced hot-summer humid continental climate according to the Köppen-Geiger climatic type [42]. During the last 10 years, the mean annual temperature was 9.16 °C and the total amount of annual precipitation was 1233 mm. Approximately 71.6% of the total annual precipitation was concentrated from June to September [43]. The research area is a mountainous highland with a very low rate of arable land with 270.17 km^2^ (89.0%) of forest land, 14.22 km^2^ (4.7%) of agricultural land, and 19.13 km^2^ (6.3%) of other areas. It mainly consists of sedimentary rocks (sandstone and limestone) and anthracite and is the region where Korea’s representative coal mines existed and produced approximately 72.9% of the national coal production.

The NHJ stream length is 29.10 km and the watershed area is 204.10 km^2^. NHJ is the main tributary of Nakdonggang River, which is the longest river in Korea and the main drinking water source in the middle eastern area of Korea. A total of 14 waste coal mines are located near the upstream of NHJ. The amount of waste produced is 697,915 m^3^, and the volume of mine wateris 3765 m^3^/day. Substrates in NHJ are acid-sulfated in a 950 m section from the minehead, and 8084 people inhabit the area near the stream. NCA has a stream length of 18.4 km, a watershed area of 61.50 km^2^, and four waste coal mines. The waste volume is 13,430 m^3^, 174 m^3^/day mined metal volume and approximately 100 m white sediment section due to the mining water. In total, 3850 local people inhabit the NCA basin, and the number of mineheads near the NCA is six. In 2010, the heavy metal concentrations of the mine water, stream water, and groundwater were officially measured, and the results showed that Pb and Fe concentrations in NCA, as well as Cd, Pb, Fe, and Mn concentrations in NHJ, exceeded the water quality limits [44].

NSJ was considered the reference stream, with a stream length of 9.4 km and a watershed area of 49.3 km^2^. It is located in a well-preserved mountainous area, with water temperature below 20 °C even in summer and dissolved oxygen of 9 mg/L. It also provides a habitat for *Brachymystax lenok*, a cold-water fish species and a second-grade endangered species according to the Ministry of Environment, Korea.

### 2.2. Ecological Data

Benthic macroinvertebrates were collected from 23 sampling sites (NHJ: 8 sites, NCA: 7, and NSJ: 8) using a Surber sampler (30 × 30 cm^2^, 300 μm mesh) at a depth of 10 cm during April 2017. This month presents normal water flow periods, thus avoiding community variability related to the season (Figure 1). Each site was sampled in a riffle area with three replicates within a 50-m reach. Collected benthic macroinvertebrates were preserved in 95% ethanol in the field, and then the solution was replaced with 70% ethanol in the laboratory. In the laboratory, specimens were sorted, counted, and identified mostly into species level under a microscope (SMZ10; Nikon, Japan). Identification was conducted based on previously published methods [45,46,47,48,49,50].

A total of 34 environmental factors were measured at the sampling sites or in the laboratory. Geographical factors were extracted using the Spatial Analyst toolbox in ArcGIS 10.6 (ESRI, Redlands, CA, USA), including altitude, slope, distance from the source, and stream order. The ratio of land cover types (e.g., urban, agriculture, forest, grassland, wetland, and bare land), was extracted in an area of 1 km-length and 200 m-wide buffer zone (i.e., 500 m up- and downstream from benthic macroinvertebrate sampling sites) [3]. Hydrological factors, substrate composition, and water quality factors, such as pH, conductivity, dissolved oxygen (DO) and turbidity were measured in each sampling site using YSI ProDSS (YSI Inc./Xylem, Rye Brook, NY, USA), whereas biological oxygen demand (BOD), ammonia-nitrogen (NH_3_-N), nitrate-nitrogen (NO_3_-N), total nitrogen (TN), phosphate-phosphorus (PO_4_-P), total phosphorus (TP), and chlorophyll-a (Chl-a) were measured in the laboratory [51]. In the case of NHJ and NCA, heavy metal concentrations (Pb, Fe, Cd, and Mn) in stream water were also measured using ICP-OES (Optima 7300DV & Avio500; Perkin-Elmer, MA, USA) and ICP-MS (NexION 300X; Perkin-Elmer, MA, USA). However, the metal concentrations did not exceed the Korean water quality limits, and thus these factors were not included in the subsequent analysis.

### 2.3. Data Analysis

We conducted four analytical approaches (Figure 2). First, as descriptive measures, species richness, abundance, and FFGs were compared between the three streams sampled. Second, network analysis based on species co-occurrence was computed to evaluate the community structure complexity among the streams. Pairwise correlations for each species were calculated using Spearman’s correlation rank [43,52]. The number of nodes (number of vertices (species) in a network), number of edges (number of connections in a network), average node degree (average number of connections per node), average path length (expected distance between two nodes), transitivity (the tendency of the nodes to cluster together), and edge density (the values dividing existing edges dividing into all the possible edges) were calculated to compare the degree of community structure complexity [36,38]. Network analysis was conducted using the package igraph [53] in R software [54].

Third, a self-organizing map (SOM) [40] was used to characterize the sites affected by abandoned mining areas and the patterns of benthic macroinvertebrate community compositions. The SOM is composed of an input and an output layer. The input layer (74 species and 23 sites) was condensed and arrayed into a two-dimensional grid (output layer) for easy interpretation of the community. We determined the number of output units based on the formula recommended by [55]: 5× sqrt (number of samples). Thus, we used 30 output units (N = 5 × 6). The SOM units were classified by K-means cluster with the Davies-Bouldin index, which finds optimal clustering based on a partition that minimizes distances within and maximizes distances between clusters [56,57,58]. For the SOM analysis, species abundance data were log-transformed (log (x + 1)), and rare taxa (i.e., the species that occurred only once) were excluded. SOM analysis was conducted using the SOM toolbox in MATLAB version 6.1 [59]. To test significant differences in the community composition among the clusters defined by SOM analysis, we computed the multi-response permutation procedure (MRPP) using the function “mrpp” in the vegan package in R [60]. In addition, the Kruskal-Wallis (KW) test was used to compare the environmental factors and community indices among the clusters. In the cases when the environmental factors or community indices were significantly different among clusters, the nonparametric Dunn’s multiple comparison test was applied for posthoc comparisons. Kruskal-Wallis and Dunn’s multiple comparison tests were conducted using the package agricolae [61] in R.

Finally, an indicator species analysis was applied to identify the species (or taxa) representing each cluster [62,63,64]. Indicator species were determined based on the relative abundance and relative occurrence frequency among the defined clusters. The Indicator Value (IndVal) ranges from 0 to 100 (i.e., all the individuals in a certain species are included only in a certain cluster) [38]. A species (or taxa) is determined as an indicator species only if IndVal for a particular cluster is significant and higher than 25% (*p* < 0.05) [62]. Indicator species analysis was performed using the “indval” function in the Labdsv package [65] in R.

## 3. Results

### 3.1. Differences in Benthic Macroinvertebrate Community among Streams

The average number of species was the highest between the sites in the NSJ stream (reference stream) with 32 species (five phyla, seven classes, 13 orders, 42 families, and 76 species in total), followed by NHJ (four phyla, six classes, 13 orders, 37 families, and 55 species) and NCA (five phyla, seven classes, 14 orders, 33 families, and 49 species) with 16 species. In the NHJ stream, species richness decreased from eight species in site 2 to two in site 4, where stream sediments consisted of acid sulfate soils but increased from 14 species in site 5 to 30 species in site 8 (the highest number of species in NHJ) (Figure 3). In the NCA stream, only six species were observed at site 2. In the NSJ stream, species richness ranged from 24 (site 7) to 40 (site 2).

Abundance also decreased from site 2 (1237 individual/m^2^) to site 4 (7) in the NHJ stream, similar to the pattern of species richness. In the NCA stream, abundance was the lowest at site 2 (219). In the NSJ stream, abundance ranged from 3270 individual/m^2^ (site 4) to 11,270 individual/m^2^ (site 8).

For FFGs, regardless of the mining impacts, both the abundance and species richness of collector gatherers (CG) were the highest in all streams. However, the percentage of CG was much higher in NCA (SR: 38.1%, abundance: 87.4%) and NHJ (SR: 42.4%, abundance: 88.2%) than in NSJ (SR: 25.2%, abundance: 59.9%) (Figure 4). In the case of scrapers (SC), both the species richness and abundance were higher in NSJ (SR: 33.1%, abundance: 20.1%) than in NCA (SR: 26.9%, abundance: 5.5%) and NHJ (SR: 19.0%, abundance: 1.1%).

Considering the differences between study sites in each stream, the ratio of SC decreased from site 1 (26.3% in abundance and 1.4% in species richness) to site 4 (0.0%) in NHJ. SC and collector-filterers were not observed in site 2 in NCA. The ratio of shredders (0.0–31.1% in abundance and 0.0–16.7% in species richness) and predators decreased, whereas that of SC increased from up- to downstream in NSJ.

### 3.2. Species Co-Occurrence Patterns of the Benthic Macroinvertebrate Community

Co-occurrence networks indicated a clear difference between the streams affected by the abandoned mining area and the reference stream. For example, the networks were less connected in the NHJ and NCA streams than in the NSJ stream (Table 1 and Figure 5). In addition, the average node degree and number of edges were highest in sites from the NSJ stream (38.68 and 1635), followed by those in the NHJ (25.60 and 826) and NCA (22.94 and 622) streams.

### 3.3. Patterns of Benthic Macroinvertebrate Community

The benthic macroinvertebrate communities were patterned and classified using the SOM learning process based on the similarities of the community composition (Figure 6). The SOM classified output units into five clusters (1–5) based on the K-means cluster with the Davies-Bouldin index. Benthic macroinvertebrate communities among five clusters were significantly different according to MRPP (A = 0.30, *p* < 0.001). These five clusters represented the differences of the community according to the gradients of mining impacts. Samples from the NSJ stream were all included in clusters 1 (site 1–4) and 5 (site 5–8); samples from sites located near the mining water inflow area were mainly included in cluster 4; samples from sites where stream sediments were mainly acid-sulfated were included in cluster 2; samples from sites that had recovered from mining impacts were included in cluster 3 in NHJ and NCA streams.

Community indices and environmental factors also differed between the five clusters (Table 2, Table 3 and Table 4). For example, species richness (KW = 18.5, *p* < 0.05) and Plecoptera species richness (KW = 16.5, *p* < 0.05) were higher in clusters 1 and 5. In addition, the Shannon diversity index (KW = 16.4, *p* < 0.05), Ephemeroptera, Plecoptera, and Trichoptera (EPT) richness (KW = 18.7, *p* < 0.05), and EPT abundance (KW = 16.3, *p* < 0.05) were higher in clusters 1, 3, and 5. In the FFGs, the ratio of SC (abundance: KW = 15.3, *p* < 0.05; SR: KW = 13.0, *p* < 0.05) was higher in cluster 1 (abundance: 17.4% and SR: 26.6%) and 5 (abundance: 66.2% and SR: 28.0). Regarding environmental factors, the ratio of forest (KW = 11.6, *p* < 0.05) in land use was higher in clusters 1 (92.3%) and 4 (93.8%), although all the clusters showed a high ratio of forest (more than 55% on average). There was no significant difference in substrate composition, except for the size <0.063 mm (KW = 12.6, *p* < 0.05). In terms of water quality, conductivity (KW = 15.3, *p* < 0.05) was higher in clusters 2 (360.8 μS/cm), 3 (379.4 μS/cm), and 4 (370.3 μS/cm), respectively. The concentration of T-N (KW = 17.6, *p* < 0.05) was higher in clusters 2 (2.11 mg/L) and 3 (2.1 mg/L).

### 3.4. Indicator Species

Forty taxa were selected as indicator species for the five clusters from the SOM analysis based on IndVal (*p* < 0.05) (Table 5). In cluster 1, indicator species were mainly included in the EPT taxa. In particular, the species in Plecoptera, such as *Kamimuria coreana*, *Stavsolus japonicas*, *Rhopalopsole mahunkai*, and *Sweltsa nikkoensis* were selected in cluster 1. In cluster 5, 17 species were selected in total, and among them, species such as *Ecdyonurus kibunensis* and *Hydropsyche orientalis*, which generally inhabit mid- to downstream, were included. In clusters 2 and 4, only one (*Physa acuta*) and two species (*Tipula* KUa and *Nemoura* KUb) were selected. In cluster 3, 13 species were selected, including Ephemeroptera and Trichoptera species, such as *E. levis*, *Epeorus pellucidus*, *Hydropsyche kozhantschikovi*, and *Hydroptila* KUa.

## 4. Discussion

Our study showed that the community complexity of benthic macroinvertebrates was lower in streams located near mining areas than in the reference stream. For example, the biodiversity of benthic macroinvertebrates decreased in the study sites (site 2 to 4), less than 1.8 km away from a site to which effluents of mining area directly flow, and the biodiversity was recovered starting from 4.1 km downstream of the effluent site (site 5) in NHJ. Although the effects of the effluents may vary depending on the amount of pollutants and the type of mining industry, the difference in the community and diversity of benthic macroinvertebrates between the study sites reflected the degree of proximity to the mining area (influence zone).

In this study, based on both physicochemical and biological parameters, NHJ and NCA were divided into three sections: the sites not affected by mining (the upper sections in both streams), sites directly affected by mining (e.g., site 2 to 4 in NHJ), and sites that were recovered from mining impacts (e.g., site 5 to 8 in NHJ). In the case of the least-disturbed stream, NSJ was largely divided into two areas: the upper-middle stream (site 1 to 4) and middle-downstream (site 5 to 8).

The co-occurrence networks revealed that the stability (complexity) of the benthic macroinvertebrate community was lower in the streams located near mining areas than in the reference stream. Network size and connectivity between species were lower in NHJ and NCA than in NSJ, resulting in the loss of biodiversity and biotic integrity [66,67]. Moreover, the average path length representing the expected distance between the two species was higher in NHJ and NCA than in NSJ.

These three streams, as typical mountainous streams, have similar environmental conditions except for the acid mining impacts: high altitude (from 514 to 898 m), high forest ratio (from 44 to 100%) in land use, and a high proportion of large substrate (i.e., cobble and boulder) in the streambed. Among these streams, the major differences were the anthropogenic impacts. In particular, the study sites, where the whitening/reddening of stream sediment was directly observed, showed relatively high values of conductivity, T-N, and T-P, which was caused by the effluent of treated wastewater from mining areas.

The species, such as chironomids, which have short life cycles (1 month) to endure their life in unstable and/or disturbed habitats, could be dominant in NHJ and NCA. In addition, SC, which scrape algae and organic matter from the surface of rocks and stream plants, were not observed in the FFG. According to the database of the National Aquatic Ecological Monitoring Program of South Korea [68], the diversity and ash-free dry mass (the weight of periphyton per unit area, cm^3^) of periphyton, which is the major food source of SC, was much lower in NHJ and NCA than in NSJ. Owing to a lower quality and amount of food sources, as well as deterioration of water quality, no SC was observed in the sites affected by mining effluents. Our results were similar to those of other studies [69,70,71,72,73,74] that showed lower species richness and abundance of SC owing to the increased osmoregulation stress [75,76,77] and heavy metal bioaccumulation due to heavy metal contaminated biofilm, which is the main food source of SC [21,78,79,80]. The loss of certain FFGs and/or taxa of benthic macroinvertebrates is critical in freshwater ecosystems because each of them is responsible for certain roles, such as food sources of higher trophic organisms, nutrient cycling, leaf litter decomposition and/or bioremediation from disturbed habitats [81,82,83].

Furthermore, some of Ephemeroptera (e.g., *Baetis*) and Trichoptera (e.g., *Hydropsyche* and *Polycentropus*) groups are relatively tolerant to mining impacts [7,84,85]. For example, *B. rhodani* was relatively abundant in the Nent River, where high concentrations of Zn were continuously observed even though mining ceased at the beginning of the 1900s. Non-cased Trichoptera are tolerant to moderately polluted areas (<500 mg Zn/L) [13]. Although these two taxa were not included as indicator species in sites 2 and 4, which were directly affected by effluents, *Baetid* groups such as *Baetis fuscatus* and *Baetis ursinus*, and Trichoptera species such as *Hydropsyche valvata*, *Cheumatopsyche* KUa, and *Hydropsyche kozhantschikovi*, were selected as indicator species in the recovered sites (cluster 3 in SOM). Furthermore, the abundance and species richness of SC, such as *Ecdyonurus levis*, *Epeorus pellucidus*, and *Semisulcospira libertine*, were significantly higher in the recovered sites.

Our results evaluated the impact range of the mining area and the differences in community complexity between two streams located near mining areas and one stream in the least-disturbed area. However, a high-efficiency mining water treatment plant was recently constructed (3000 m^3^ of wastewater per day) near site 2 in NHJ in 2019; therefore, long-term monitoring should be performed to assess the recovery of the macroinvertebrate community from the mining impacts and to further understand the recovery processes and periods of freshwater ecosystem. The community recovery of benthic macroinvertebrates after the remediation or restoration treatment can be different according to the degree of contamination, the removal of metal contaminated soils, the closeness of upstream sources of colonization, hydrologic conditions, the effectiveness of remediation, etc. [86]. For example, in the upper Arkansas River near mining area in Colorado, USA, after mine drainage treatment, macroinvertebrate community became similar with the community reference sites (upstream of mining-impacted area) within two years, and especially increased EPT taxa [20]. In contrast, 20–29 years of long-term monitoring in four mining-impacted watersheds in the western USA revealed that benthic macroinvertebrate species richness increased within 10.25 years on average after remediation activity from mining, including water treatment, construction of contaminated ponds, revegetation of riparian areas [35]. In the Nent catchment, where active mining ceased in the 1900s, high concentrations of Zn are still detected even 100 years later, causing the benthic macroinvertebrate community to remain low compared with that of unpolluted tributaries [13].

In 1 year survey, network analysis and SOM were used to quantify the area influenced by the mining activities which needs to be managed to enhance the benthic macroinvertebrate diversity. These two analytical methods can be further applied to evaluate whether the diversity of benthic macroinvertebrates will recover after the completion of mining treatment. For instance, SOM can be used to estimate the time required for the benthic macroinvertebrate community near mining area to become similar to that of the reference sites and to evaluate if its stability (or complexity) increases after the completion of mining treatment using network analysis (e.g., comparison of the number of edges and average node degree).

## 5. Conclusions

This study demonstrates that the benthic macroinvertebrate community was less diverse and complex in streams located near mining areas than in the reference stream, with different functions (e.g., no scrapers in the stream nearly located in a mining area) and structures (e.g., the lowest species richness). Our approaches of network analysis and SOM could provide analytical methods for quantifying the impacts of mining activities on the benthic macroinvertebrate community. We revealed the endpoints of the impacts of the mining area (e.g., 4.1 km downstream of the effluent site in NHJ) and a reduced interaction among benthic macroinvertebrates in mining-impacted areas. We showed that those two analytical methods are useful to quantify the area influenced by the mining, which should be prioritized for establishing and implementing conservation and management plans to enhance the community diversity of benthic macroinvertebrates.

## Figures and Tables

**Figure 1 ijerph-18-11132-f001:**
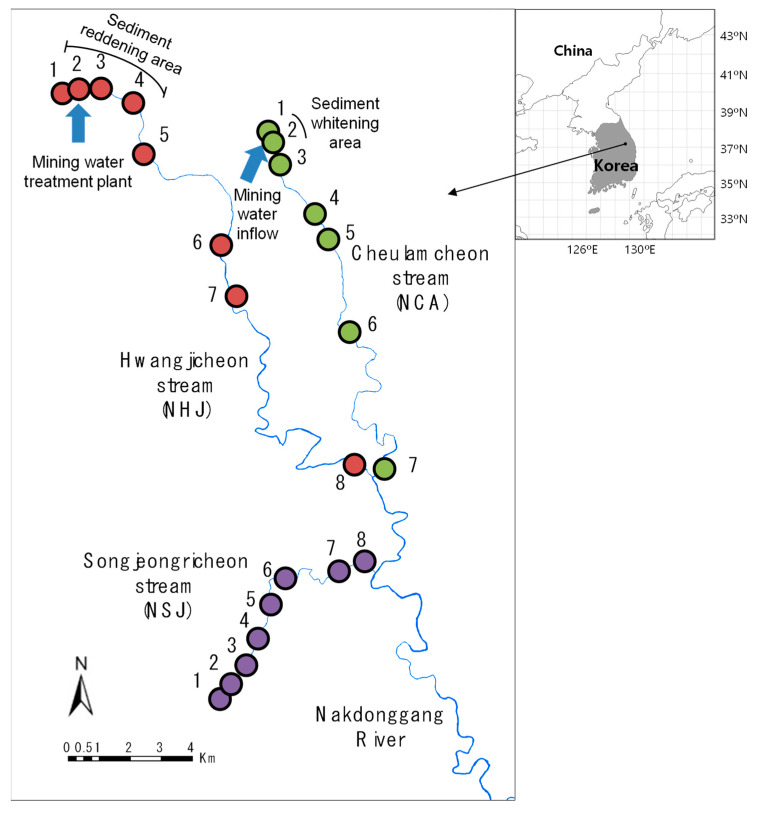
Sampling sites in three streams. Numbers 1 to 8 in each stream indicate sampling sites.

**Figure 2 ijerph-18-11132-f002:**
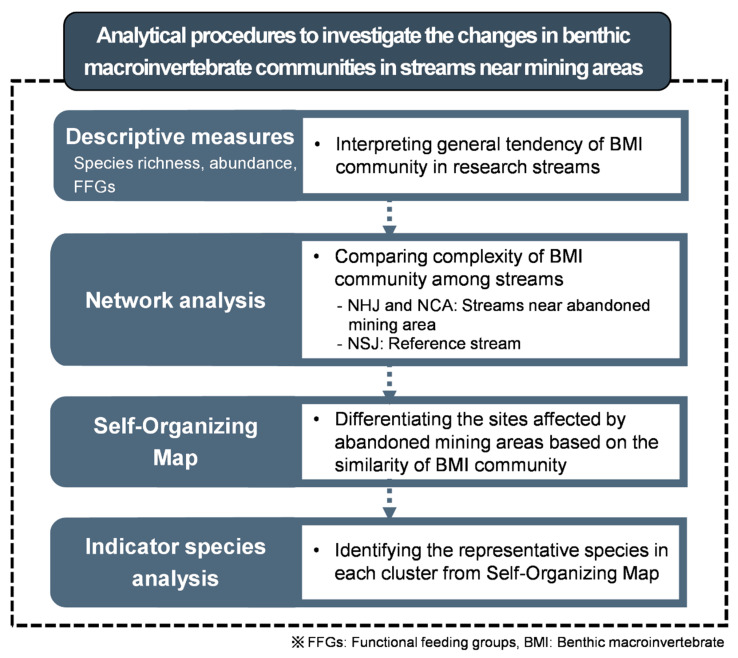
The procedure for data analysis.

**Figure 3 ijerph-18-11132-f003:**
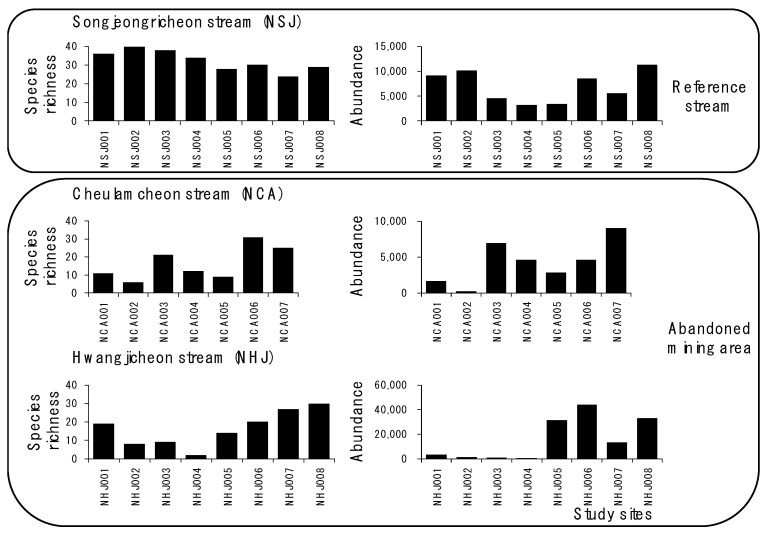
Differences in species richness and abundance in streams in the reference site (NSJ) and abandoned mining areas (NCA and NHJ).

**Figure 4 ijerph-18-11132-f004:**
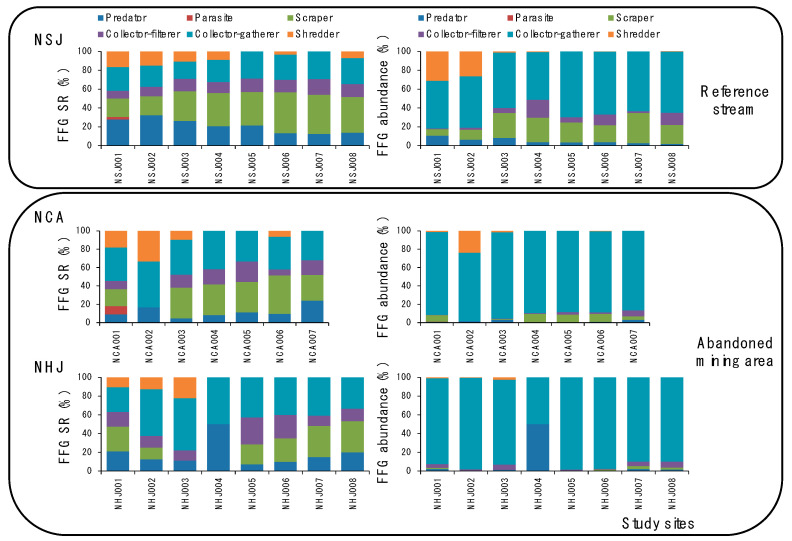
Differences in functional feeding group (FFG, %) based on abundance and species richness (SR) in streams in reference sites and abandoned mining area (NSJ: Songjeonrichoen stream, NCA: Cheulamcheon stream and NHJ: Hwangjicheon stream).

**Figure 5 ijerph-18-11132-f005:**
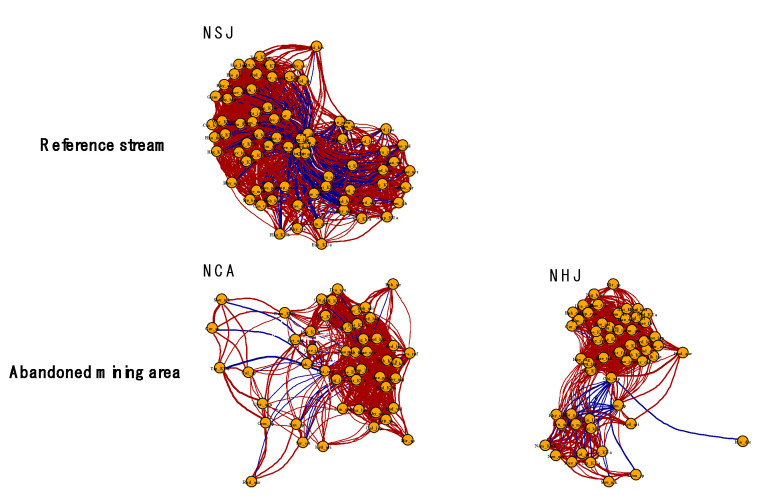
Co-occurrence networks among benthic macroinvertebrate species from streams located near abandoned mining area (NCA and NHJ) and reference stream (NSJ). Nodes indicates each species observed in each stream, and lines represent connections among species.

**Figure 6 ijerph-18-11132-f006:**
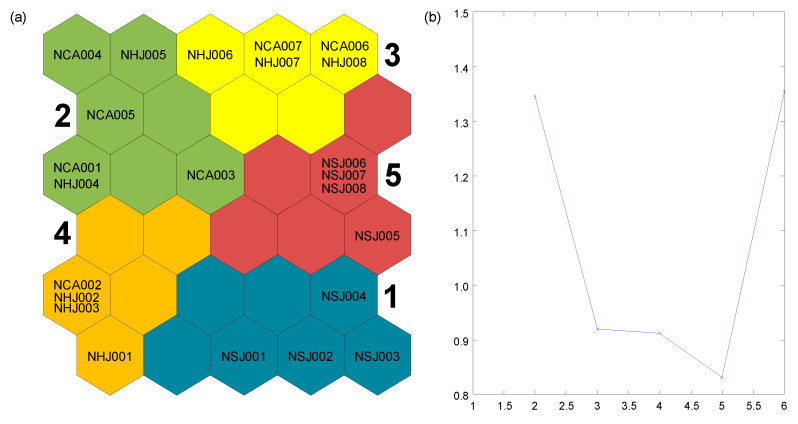
(**a**) Distribution and classification of sampling sites in the self-organizing map based on the abundance of benthic macroinvertebrates, (**b**) K-means cluster with Davies-Bouldin index. Numbers 1 to 5 indicate cluster defined by self-organizing map.

**Table 1 ijerph-18-11132-t001:** Parameters of the benthic macroinvertebrate co-occurrence network (NSJ: Songjeonrichoen stream, NCA: Cheulamcheon stream, and NHJ: Hwangjicheon stream).

Parameters	Reference Stream	Streams Located Near Abandoned Mining Areas	
	NSJ	NCA	NHJ
Number of nodes	76	49	55
Number of edges	1635	622	826
Average node degree	38.68	22.94	25.60
Average path length	1.48	1.52	1.53
Transitivity	0.70	0.73	0.78
Edge density	0.52	0.48	0.47

**Table 2 ijerph-18-11132-t002:** Differences in community indices between five clusters in a self-organizing map.

Clusters	1	2	3	4	5
Species richness	37 (3) ^a^	12 (6) ^b^	27 (4) ^ab^	11 (6) ^b^	28 (3) ^a^
Abundance	6815 (3391)	7917 (11,782)	20,819 (16,946)	1394 (1324)	7193 (3431)
Ephemeroptera	1404 (387) ^a^	99 (157) ^b^	1181 (513) ^a^	238 (442) ^ab^	2921 (1747) ^a^
Plecoptera	1324 (1444) ^a^	4 (6) ^b^	2 (5) ^b^	20 (19) ^a^	7 (5) ^ab^
Trichoptera	357 (242)	120 (191)	957 (933)	45 (88)	676 (596)
Ephemeroptera	11 (2) ^a^	2 (2.3) ^b^	9.2 (2.9) ^ab^	1.5 (1.7) ^b^	12 (0.8) ^a^
Plecoptera	6.3 (0.5) ^a^	0.5 (0.5) ^b^	0.2 (0.4) ^b^	1.3 (0.5) ^ab^	1.3 (0.5) ^a^
Trichoptera	8.8 (1.0) ^a^	2.5 (2.6) ^b^	7.2 (1.1) ^a^	2 (3.4) ^b^	6.5 (1.9) ^ab^
EPT abundance	3085 (1258) ^a^	223 (348) ^b^	2141 (925) ^a^	304 (534) ^ab^	3605 (2267) ^a^
ETP richness	26 (2.2) ^a^	5 (4.9) ^b^	16.6 (3.7) ^a^	4.8 (4.9) ^ab^	19.8 (1.7) ^a^
Non-insecta abundance	452 (619) ^a^	185 (165) ^ab^	456 (280) ^a^	21 (21) ^b^	150 (70) ^a^
Other-insecta abundance	3278 (1916) ^ab^	7509 (11,494) ^a^	18,222 (16,269) ^a^	1069 (827) ^b^	3438 (1207) ^a^
Non-insecta species richness	3 (0.8) ^ab^	4 (2.4) ^a^	5.6 (0.9) ^a^	1.5 (0.6) ^b^	3 (1.2) ^a^
Other-insecta species richness	8 (2.2) ^a^	2.5 (0.8) ^b^	4.4 (1.5) ^ab^	4.3 (1) ^a^	5 (0.8) ^a^
Dominance index	0.61 (0.09) ^b^	0.95 (0.04) ^a^	0.82 (0.1) ^ab^	0.89 (0.06) ^a^	0.64 (0.06) ^b^
Shannon diversity	3.06 (0.34) ^a^	0.78 (0.28) ^b^	1.56 (0.71) ^a^	1.14 (0.63) ^ab^	2.69 (0.28) ^a^
Richness index	17.71 (1.12) ^a^	27.93 (24.07) ^a^	16.12 (1.57) ^b^	23.03 (4.04) ^a^	17.58 (1.06) ^ab^
Evenness	0.42 (0.05) ^a^	0.11 (0.04) ^b^	0.21 (0.1) ^a^	0.16 (0.09) ^ab^	0.37 (0.04) ^a^

Values represent mean (standard deviation); Lowercase letters represent significant differences in environmental values among the five clusters based on the Kruskal-Wallis test and Dunn’s multiple comparison test (*p* < 0.05).

**Table 3 ijerph-18-11132-t003:** Differences in functional feeding groups between five clusters in the self-organizing map.

Category	Cluster	1	2	3	4	5
FFG abundance	Predator (%)	7.3 (2.9)	9.1 (20.1)	1.6 (1)	1.4 (0.8)	3 (0.9)
	Parasite (%)	0 (0)	0 (0.1)	0 (0)	0 (0)	0 (0)
	Scraper (%)	17.4 (10.2) ^a^	4.4 (4.5) ^ab^	3.9 (2.9) ^a^	0.4 (0.6) ^b^	22.8 (6.2) ^a^
	Collector filterer (%)	6.8 (8.3)	1 (1)	4 (2.8)	2.7 (2.6)	7.9 (5.1)
	Collector gatherer (%)	53.6 (4) ^b^	85.1 (17.6) ^a^	90.5 (4.2) ^a^	88.6 (9.9) ^a^	66.2 (2.7) ^ab^
	Shredder (%)	14.8 (16.2)	0.5 (0.7)	0.1 (0.1)	6.9 (11.3)	0 (0.1)
FFG species richness	Predator (%)	26.8 (4.9) ^a^	15.1 (17.2) ^b^	15.7 (6.3) ^ab^	15.3 (4.5) ^a^	15.3 (4.1) ^a^
	Parasite (%)	0.7 (1.4)	1.5 (3.7)	0.0 (0.0)	0.0 (0.0)	0.0 (0.0)
	Scraper (%)	26.6 (8.1) ^a^	23.3 (13.2) ^ab^	32.3 (6.5) ^a^	9.7 (12.5) ^b^	39.7 (3.5) ^a^
	Collector filterer (%)	10.8 (2.1)	15.1 (10)	14.4 (6.9)	9.9 (6.9)	14.5 (1.5)
	Collector gatherer (%)	22.4 (2.8) ^b^	40.4 (5.9) ^a^	36.3 (3.9) ^a^	45.5 (13) ^a^	28 (1.1) ^ab^
	Shredder (%)	12.8 (3.7)	4.6 (7.7)	1.3 (2.9)	19.6 (10.5)	2.6 (3.3)

Values represent mean (standard deviation); Lowercase letters represent significant differences of environmental values among the five clusters based on the Kruskal-Wallis test and Dunn’s multiple comparison test (*p* < 0.05).

**Table 4 ijerph-18-11132-t004:** Differences in environmental factors between five clusters in self-organizing map.

Category	Cluster	1	2	3	4	5
Geological factors	Altitude	738 (85) ^ab^	741 (66) ^a^	606 (61) ^b^	857 (47) ^a^	556 (42) ^b^
	DFS (km)	0.97 (0.36) ^b^	1.39 (0.51) ^ab^	2.76 (0.47) ^a^	0.85 (0.15) ^b^	2.06 (0.33) ^a^
	Slope	8.27 (2.1)	9.48 (3.37)	15.11 (6.17)	11.03 (7.96)	12.15 (8.73)
	Stream order	2 (1) ^b^	3 (1) ^ab^	5 (1) ^a^	2 (0) ^b^	5 (1) ^a^
Hydrology	Depth (cm)	9.8 (3.2)	11.4 (2.4)	16.3 (4.9)	10.8 (4.1)	19.5 (10.8)
	Velocity (cm/s)	0.53 (0.18)	0.3 (0.16)	0.59 (0.15)	0.49 (0.21)	0.59 (0.26)
Land use (%)	Urban	2.9 (3.8)	6.1 (6.4)	22.4 (17.1)	3.2 (2.9)	4.1 (2.9)
	Agriculture	4.8 (5.8)	10.5 (9.2)	12.8 (8.7)	0.6 (0.8)	16.7 (16.2)
	Forest	92.3 (9.4) ^a^	77.6 (16.0) ^a^	55.1 (14.3) ^b^	93.8 (7.6) ^a^	76.2 (19.2) ^ab^
	Grassland	0 (0)	1.2 (2.4)	0.9 (1)	1.8 (3.4)	0.8 (1.1)
	Wetland	0 (0)	0 (0)	2.3 (3.1)	0 (0)	0.1 (0.1)
	Bare land	0 (0) ^b^	4.6 (6.8) ^a^	6.7 (7.1) ^a^	0.6 (0.9) ^ab^	2.1 (1.5) ^a^
Substrate composition (%)	<0.063 mm	0.0 (0.0) ^ab^	0.2 (0.2) ^a^	0.4 (0.3) ^a^	0.5 (0.5) ^a^	0.0 (0.0) ^b^
	0.063–2 mm	0.9 (0.2)	0.5 (0.4)	0.7 (0.2)	0.9 (0.2)	1.1 (0.4)
	2–4 mm	1.1 (0.3)	0.9 (0.1)	0.9 (0.2)	0.9 (0.1)	1.2 (0.5)
	4–8 mm	1.4 (0.2)	1.3 (0.6)	1 (0.2)	1.1 (0.4)	0.9 (0.8)
	8–16 mm	1.8 (0.7)	2.3 (1.6)	1.9 (0.2)	2 (0.4)	1.2 (0.8)
	16–32 mm	4 (0.6)	4.4 (3.2)	3.2 (0.8)	4.2 (1.2)	2 (1.8)
	32–64 mm	7.9 (2.6)	7.9 (4.9)	6 (2.4)	10.1 (4.9)	6.5 (3.2)
	64–128 mm	22.1 (4.4)	25.5 (9.2)	18 (6.2)	21.6 (8.5)	23.7 (4.6)
	128–256 mm	55.4 (7.3)	56.9 (15)	65.4 (9.4)	56.1 (12.5)	52.8 (12.1)
	>256 mm	5.4 (9.3)	0 (0)	2.5 (4.2)	2.5 (3.3)	10.7 (21.4)
Water quality	DO (%)	85.9 (1.8) ^b^	100.2 (10.2) ^a^	108.2 (11) ^a^	88.8 (0.4) ^a^	88.1 (2.2) ^ab^
	DO (mg/L)	9.59 (0.19)	10.63 (1.43)	10.73 (1.44)	9.82 (0.21)	9.35 (0.1)
	pH	7.57 (0.46) ^b^	8.35 (0.5) ^a^	8.73 (0.47) ^a^	7.73 (0.27) ^ab^	8.51 (0.23) ^a^
	Conductivity (μS/cm)	116.0 (68.8) ^ab^	360.8 (11.37) ^a^	379.4 (112.8) ^a^	370.3 (188.1) ^a^	92.3 (23.3) ^b^
	Turbidity (NTU)	1.1 (0.6)	1 (1.5)	0.2 (0.3)	1.1 (1.1)	1 (0.2)
	BOD (mg/L)	1.13 (0.24)	0.97 (0.21)	1.16 (0.29)	0.7 (0.08)	1.03 (0.17)
	NH_3_-N (mg/L)	0.01 (0.003)	0.016 (0.019)	0.013 (0.009)	0.013 (0.004)	0.008 (0.003)
	NO_3_-N (mg/L)	0.79 (0.25) ^b^	1.54 (0.41) ^a^	1.18 (0.32) ^ab^	0.71 (0.28) ^b^	1.22 (0.13) ^a^
	T-N (mg/L)	1.07 (0.36) ^b^	2.11 (0.47) ^a^	2.1 (0.11) ^a^	0.81 (0.28) ^b^	1.48 (0.13) ^ab^
	PO_4_-P (mg/L)	0.003 (0)	0.008 (0.007)	0.006 (0.004)	0.003 (0.001)	0.003 (0)
	T-P (mg/L)	0.008 (0.002)	0.013 (0.009)	0.015 (0.008)	0.006 (0.001)	0.005 (0)
	Chl-a (mg/L)	0.53 (0.13)	0.77 (0.53)	0.5 (0.07)	0.48 (0.05)	0.53 (0.05)

Values represent mean (standard deviation); Lowercase letters represent significant differences of environmental values among the five clusters based on the Kruskal-Wallis test and Dunn’s multiple comparison test (*p* < 0.05).

**Table 5 ijerph-18-11132-t005:** Indicator species (or taxa) for five clusters from the self-organizing map.

Cluster	Order	Family	Species (Taxa)	IndVal	*p* Value
1	Ephemeroptera	Heptageniidae	*Cinygmula* KUa	0.75	0.009
1	Ephemeroptera	Heptageniidae	*Epeorus curvatulus*	0.64	0.007
1	Ephemeroptera	Ephemerellidae	*Cincticostella levanidovae*	0.48	0.014
1	Plecoptera	Nemouridae	*Nemoura* KUa	1.00	0.002
1	Plecoptera	Perlodidae	*Stavsolus japonicus*	1.00	0.001
1	Plecoptera	Perlidae	*Kamimuria coreana*	1.00	0.001
1	Plecoptera	Nemouridae	*Amphinemura coreana*	0.75	0.006
1	Plecoptera	Leuctridae	*Rhopalopsole mahunkai*	0.75	0.01
1	Plecoptera	Chloroperlidae	*Sweltsa nikkoensis*	0.67	0.002
1	Trichoptera	Philopotamidae	*Wormaldia* KUa	0.75	0.005
1	Trichoptera	Hydrobiosidae	*Apsilochorema* KUa	0.62	0.021
1	Trichoptera	Rhyacophilidae	*Rhyacophila shikotsuensis*	0.56	0.032
1	Trichoptera	Lepidostomatidae	*Lepidostoma* KUb	0.47	0.044
1	Coleoptera	Elmidae	Elmidae sp.	0.80	0.004
1	Diptera	Tipulidae	*Hexatoma* KUa	0.64	0.008
1	Diptera	Ceratopogonidae	Ceratopogonidae sp.	0.62	0.006
1	Diptera	Tipulidae	*Dicranota* KUa	0.53	0.018
2	Basomatophomra	Physidae	*Physa acuta*	0.41	0.039
3	Archioligochaeta	Naididae	*Chaetogaster limnaei*	0.72	0.003
3	Archioligochaeta	Tubificidae	*Limnodrilus gotoi*	0.48	0.003
3	Mesogastropoda	Pleuroceridae	*Semisulcospira libertine*	0.41	0.046
3	Ephemeroptera	Heptageniidae	*Ecdyonurus levis*	0.79	0.001
3	Ephemeroptera	Heptageniidae	*Epeorus pellucidus*	0.43	0.037
3	Ephemeroptera	Baetidae	*Baetis fuscatus*	0.80	0.001
3	Ephemeroptera	Baetidae	*Baetis ursinus*	0.61	0.001
3	Trichoptera	Hydroptilidae	*Hydroptila* KUa	0.87	0.001
3	Trichoptera	Hydropsychidae	*Hydropsyche valvata*	0.71	0.002
3	Trichoptera	Hydropsychidae	*Cheumatopsyche* KUa	0.41	0.019
3	Trichoptera	Hydropsychidae	*Hydropsyche kozhantschikovi*	0.40	0.04
3	Diptera	Tipulidae	*Antocha* KUa	0.39	0.002
3	Diptera	Chironomidae	Chironomidae spp	0.24	0.033
4	Plecoptera	Nemouridae	*Nemoura* KUb	0.44	0.041
4	Diptera	Tipulidae	*Tipula* KUa	0.65	0.007
5	Ephemeroptera	Heptageniidae	*Ecdyonurus bajkovae*	0.71	0.002
5	Ephemeroptera	Heptageniidae	*Ecdyonurus kibunensis*	0.47	0.012
5	Ephemeroptera	Ephemerellidae	*Uracanthella rufa*	0.57	0.004
5	Ephemeroptera	Leptophlebiidae	*Paraleptophlebia chocorata*	0.51	0.012
5	Ephemeroptera	Ephemeridae	*Ephemera strigata*	0.48	0.05
5	Trichoptera	Glossosomatidae	*Glossosoma* KUa	0.60	0.006
5	Trichoptera	Hydropsychidae	*Hydropsyche orientalis*	0.48	0.018

Only taxa with significant values are shown. IndVal: indicator value.

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
