# Peer review of "Evaluation of the Impacts of Abandoned Mining Areas: A Case Study with Benthic Macroinvertebrate Assemblages"

_ijerph, 2021, doi:10.3390/ijerph182111132_

Round 1

Reviewer 1 Report

The manuscript shows a valuable study on the mining activity influencing the natural habitat. The content has a publishing potential however need quite deep revision and improvement. Starting with the introduction part, more insight on the issue and a more broad explanation of the purpose of the study should be involved. The weak part is the material and methods section, ecological data, data analyses and the classification need much wider explanation all the detail on the approach should be explained in more detail. The results section is very broad but difficult to follow since the full explanation of the methodology has not been provided. Another issue is the balance between each section that needs improving, please focus on materials and analyses more (presenting a diagram explaining the study approach would be useful). When it comes to conclusions the critical review of short term monitoring results is appreciated, but this raised a question of how reliable are the research findings. Clearly, the discussion needs to be extended proving the finding are promissing.

Author Response

The manuscript shows a valuable study on the mining activity influencing the natural habitat. The content has a publishing potential however need quite deep revision and improvement. Starting with the introduction part, more insight on the issue and a more broad explanation of the purpose of the study should be involved. The weak part is the material and methods section, ecological data, data analyses and the classification need much wider explanation all the detail on the approach should be explained in more detail. The results section is very broad but difficult to follow since the full explanation of the methodology has not been provided. Another issue is the balance between each section that needs improving, please focus on materials and analyses more (presenting a diagram explaining the study approach would be useful). When it comes to conclusions the critical review of short term monitoring results is appreciated, but this raised a question of how reliable are the research findings. Clearly, the discussion needs to be extended proving the finding are promissing.

→ Thank you for your comments. We revised our manuscript based on your comments. We mainly revised discussion section to suggest the promising finding of our research such as the quantification of disturbed area using two modelling approaches.

→ We surveyed sampling when the impacts of heavy rain or flooding were minimum in order to reflect the general status of benthic macroinvertebrates in spite of one year sampling. We can clearly quantify the disturbed area to be managed based on self-organizing map and network analysis. In addition, because our sampling was conducted before completion of mining water treatment plants, long-term monitoring is necessary to check whether or not the diversity of benthic macroinvertebrate is recovered.

→ We also revised material and methods section such as adding the geological and mineralogical characteristics of the mining areas, climate conditions, etc. In addition, we also added explanation of unfamiliar terminology from two modelling analyses and analytical diagram.

Reviewer 2 Report

The work indicates that the benthic macroinvertebrate community is highly varied and is modified by the mining environment. It is of great interest and of environmental change

 It must be corrected:

Line 98: Indicate the geological and mineralogical characteristics of the mining areas

Figure 1: include in the figure, the mining areas and their area of influence

Line 116: Indicate the Köppen-Geiger climatic type

Table 4: Indicate EC units and all the water quality parameters.

Author Response

The work indicates that the benthic macroinvertebrate community is highly varied and is modified by the mining environment. It is of great interest and of environmental change

 It must be corrected:

Line 98: Indicate the geological and mineralogical characteristics of the mining areas

→ Thank you for your comments. We added it as below (you can find it in 2.1 Study area section)

→ The research area is a mountainous highland with a very low rate of arable land with 270.17 km2 (89.0%) of forest land, 14.22 km2 (4.7%) of agricultural land, and 19.13 km2 (6.3%) of other areas. It mainly consists of sedimentary rocks (sandstone and limestone) and anthracite and is the region where Korea’s representative coal mines existed and produced approximately 72.9% of the national coal production.

Figure 1: include in the figure, the mining areas and their area of influence

→ Thank you for your comments. At first, we also wanted to add the mining areas but the location of mines does not open to the public. Only the number of mines in each province or streams are opened. So unfortunately, we could not add the mining areas. However, in our figure, we marked the influence area such as acid-sulfated and sediment whitening area in the streams near the abandoned mining area.

Line 116: Indicate the Köppen-Geiger climatic type

→ We added the climate type

Table 4: Indicate EC units and all the water quality parameters.

→ We added the units for all the water quality parameters in table 4.

Round 2

Reviewer 1 Report

The Authors took into account the comments of the reviewers in the revised version of the manuscript. The article contains valuable content, presented in a clear way. I suggest accepting the article for publication in its present form. Good luck!